# Developmental Associations between Neurovascularization and Microglia Colonization

**DOI:** 10.3390/ijms25021281

**Published:** 2024-01-20

**Authors:** G. Jean Harry

**Affiliations:** Mechanistic Toxicology Branch, Division of Translational Toxicology, National Institute Environmental Health Sciences, 111 T.W. Alexander Drive, Research Triangle Park, Durham, NC 27709, USA; harry@nih.gov

**Keywords:** vascularization, microglia colonization, development, yolk sac, tissue resident macrophages

## Abstract

The temporal and spatial pattern of microglia colonization and vascular infiltration of the nervous system implies critical associated roles in early stages of nervous system development. Adding to existing reviews that cover a broad spectrum of the various roles of microglia during brain development, the current review will focus on the developmental ontogeny and interdependency between the colonization of the nervous system with yolk sac derived macrophages and vascularization. Gaining a better understanding of the timing and the interdependency of these two processes will significantly contribute to the interpretation of data generated regarding alterations in either process during early development. Additionally, such knowledge should provide a framework for understanding the influence of the early gestational environmental and the impact of genetics, disease, disorders, or exposures on the early developing nervous system and the potential for long-term and life-time effects.

## 1. Introduction

Microglia represent the yolk sac derived tissue resident macrophage of the nervous system and as such, they have unique tasks associated with maintaining homeostatic balance of the associated organs. The explosion in published literature on the role of microglia in neurodegenerative disease, nervous system injury and repair, and aspects of neural circuitry refinement has provided a wealth of information on microglia. However, our understanding of immature microglia is less clear. This is mostly due to the relatively rapid dynamics of tissue development and the shifting location, morphology, and functional phenotype of microglia [1,2]. Recent interest and attention have been focused on microglia for their phagocytic functions that act to regulate the neuronal progenitor pool size [3] and facilitate the refinement of the neural circuitry by contributing to the normal processes of axonal elongation, synaptic refinement, and synaptic pruning [4,5,6,7]. Other work has laid the foundation for understanding how microglia interact with and contribute to the migration and maturation of other glial cells in the brain [8,9]. To add to the existing reviews that cover a broad spectrum of the various roles of microglia during brain development, the current review will focus on the developmental ontogeny and interdependency between colonization of the brain with yolk sac derived macrophages and neurovascularization.

The process of nervous system development involves a temporal and spatial sequence of critical events. This sequence begins with a thickening of the dorsal ectoderm to form the neural plate along the dorsal side of the embryo. With neurulation, the neural plate progressively matures into the neural tube for formation of the central nervous system (CNS), and the neural crest for formation of the peripheral nervous system (PNS). At this early stage, the parenchymal wall is composed of the columnar cells of the neuroepithelium. These cells undergo expansion, and then the cells divide to produce the first neurons (reviewed in [10,11]). The neuroepithelial cells then transition to radial glial (RG) cells that either symmetrically divide to maintain the progenitor cell population or asymmetrically divide to produce self-renewing RG cells and intermediate neuronal or glial progenitors [12,13,14,15].

The initiation of neurogenesis corresponds with the migration of microglial progenitors and their entry into brain parenchyma. This initial entry occurs at approximately 4.5 to 5 weeks gestation in humans and at embryonic days (ED) 9–10 in mice [16,17] and utilizes proposed routes through the ventricle walls or the meninges [18,19]. This temporal and spatial co-localization of microglia with neural cells during CNS development is associated with critical cell-cell interactions that are necessary for progenitor cell proliferation, precursor cell migration, and the subsequent differentiation to neurons or glia [20]. In the developing brain, the presence of microglia appears to be associated, in a temporal and spatial manner, with the recruitment, infiltration, and branching of the neurovascular system. This co-timing has led to the speculation regarding complex interactions between colonization and innervation during development and consideration of the possibility that immature yolk sac macrophages destined to populate the nervous system utilize the vascular network for entry. Any such interactions require a coordinated effort spanning across stages of cell development, involving cell-cell contact and various developmentally related secreted factors. The synergistic and interdependent nature of these functions is critical for brain development [21,22,23].

## 2. Ontogeny of Microglia Colonization

Microglia are present in the developing brain from the early stages of neurogenesis, when neuroepithelial cells transform into radial glia (RG). In mice, yolk sac blood islands are formed around embryonic day (ED) 7.5 [24,25,26]. This involves the migration of vascular endothelial growth factor receptor (VEGFR) positive precursors of hematopoietic and endothelial cells from the primitive streak to the proximal yolk sack. Within these blood islands, multi-lineage c-kit^+^ erythromyeloid yolk sac microglia precursor cells reside [27]. These cells enter the brain rudiment via lateral ventricles and leptomeninges of the rodent by ED 9.5 and proceed to distribute throughout the cortical wall [17,18,28,29,30]. The phenotypic specification of these unique tissue-specific resident macrophages is thought to be defined by intrinsic and local microenvironment-specific transcriptional regulators [31,32,33]. A rapid proliferation occurs between ED 10.5 and birth [27], with a continued increase in microglial cells over the first 2 postnatal weeks, followed by a decline from week 3–6, and subsequent stabilization [34]. These cells mature from the early blood island CD45^+^/c-kit^low^/CX3CR1^−^/F4/80^−^ phenotype to a CD45^+^/c-kit^−^/CX3CR1^+^/F4/80^high^ phenotype prior to assuming a mature macrophage phenotype in the neuroepithelium [27]. The migration pattern follows a rostral-to-caudal gradient for colonization [16,18,28,35,36,37]. Migration of any cell within the brain does not occur in a non-directed random manner; rather, the directionality of migration relies on chemoattractant and chemorepellent signals. The bidirectional migration of microglia through the developing mouse brain suggests that molecules expressed in both the meninges and the inner region of the cerebral wall serve as chemoattractants [38].

In humans, macrophages are detected in the blood islands of the extraembryonic yolk sac within the first 2–3 weeks gestation [39,40,41,42]. The arrival of microglia progenitors occurs during the 4th gestational week. Three routes have been described for microglia entry into the brain: the leptomeningeal route, the ventricular lumen route, and the choroid plexus route. In the telencephalic wall, microglial cells displaying a rounded amoeboid morphology are detected in the ventricular lumen and ventricular zone as early as gestational week 4.5. Microglia of an amoeboid morphology accumulate in the superficial marginal zone, possibly transferring from the pial surface. By gestational week 5.5, microglia transfer from the choroid plexus into the parenchyma of the thalamic eminence [43,44,45,46,47,48]. Migration progresses along a radial and tangential pattern towards the putative white matter, subplate, and cortical plate layers. Pial cells populate the developing cortical layer I. A second wave of migration of microglia into the white matter occurs at 12–13 gestational weeks [47,48]. Upon thickening of the cortical plate by gestational week 8, intermediate rod-shaped microglial cells appear in the marginal zone, with an increased density in the ventricular zone, subventricular zone, intermediate zone, and subplate [49,50,51]. Between the 14th and 17th gestational weeks, microglia cluster near or within the white matter at the junction between the thalamus and the internal capsule, the optic tract, and the junction between the internal capsule and the cerebral peduncle followed by clusters in the corpus callosum and the periventricular hypothalamus [47,51,52]. By gestational week 18, radial migration of microglia within the cortical plate serves to populate the gray matter. Tangential migration occurs within the intermediate zone, establishing white matter microglia. Ramified microglia have been detected in the cerebellum as early as 7.5 gestational weeks [44]. In the spinal cord, colonization of the gray and white matter occurs at gestational week 9, following a route from the meninges into the gray/white matter regions [45,53]. By 18–24 weeks gestation, microglia within the spinal cord gray matter display a primary ramified morphology, while in the white matter, a primary amoeboid morphology is observed [54].

The final stages of colonization involve the recruitment of processes to regulate cell density. Recent work by Menassa et al. [55,56] demonstrated waves of proliferation in the human brain showing the highest level occurring between gestational weeks 9 and 12, followed by a second wave between gestational weeks 13 and 16, reaching a final peak density at gestational week 20, with the resulting density remaining similar to that observed in adults. This is followed by a period of apoptotic clearance of potentially excess microglia between gestational weeks 12 and 16. These density patterns were observed in Iba-1^+^ cells, with similar but lower numbers of cells identified by the transmembrane protein microglia marker, TMEM119^+^ [55]. At these early stages of brain development, the features of microglia colonization in humans were found to be similar between males and females [56]. This contrasts with previous studies in rodents that suggested increased microglia density in postnatal males, sex-dependent morphological differences lasting into juvenile and adult life stages [57], and in a delay in transcriptional maturation in males [58].

## 3. Ontogeny of Neurovascularization

During embryogenesis, vessels are generated from vascular precursor cells through the process of vasculogenesis. This initial vascular network expands via angiogenesis, the growth and branching of existing blood vessels [59]. In a process thought to be mediated by macrophages, tip cells in nascent capillaries anastomose with each other during vascular sprouting [60,61,62]. The created vascular loops move toward the center of the neural tube, establishing a temporary plexus known as the perineural vascular plexus (PNVP) around the ventricular spaces and the spinal cord central canal. In rodents, this process starts at ED 7.5–8.5 [63] or ED 8.5–9 [64] with the recruitment of endothelial cell precursors expressing VEGFR2 kinase insert domain (KID). from the adjacent lateral plate and pre-somatic mesoderm [65]. The first peaks occur around birth [66] and drive the formation and remodeling of vessels. The PNVP transforms into the arteries and veins of the pia and the leptomeninges that ensheath the CNS [22]. At ED 9.5, vessels from the PNVP sprout to invade the parenchyma and form the intraneural vascular plexus. This is followed by branching, arborization, and migration of vascular sprouts from the pial network toward the ventricles, expressing angiogenic factors such as vascular endothelial growth factor [63,67]. They then branch to the ependyma, forming the periventricular vascular plexus [68,69]. At the time of birth, the network resembles much of its adult morphology; however, during the post-natal period, there is remodeling of the pial vasculature. This process involves pruning approximately 50% of the anastomotic connections within the arterial branches and pruning of collaterals linking major cerebral arteries [70]. In addition, the diameter of remaining arterioles is altered to accommodate the expanding cortical tissue and maintain vascular tone. The pial venular network covers the cortical surface at birth and is gradually pruned from regions intervening the major draining venules between postnatal day (PND) 7 and 14. The pial capillary anastomotic plexus, a human equivalent to the PNVP of mice, is detected by gestational week 6, separated from the cortical tissue by the external glial limiting membrane. With development, the pial capillaries perforate this membrane and invade the cerebral cortex. This process begins at gestational week 8 [71], peaks around gestational week 35 [72], and predominantly occurs via sprouting angiogenesis. The penetrating arterioles and ascending venules are present at birth and continue to be refined over the early postnatal period. The developmental process follows a caudal-cephalic gradient that commences at the myelencephalon and ascends through the metencephalon, mesencephalon, diencephalon, and telencephalon [73,74,75]. The capillary network is sparse and incomplete as compared to the adult brain, and over the first few weeks following birth, it undergoes a dramatic expansion via angiogenesis [59,76,77] followed by a stabilization of capillary density. As correlated with the lower energy demands of white matter over gray matter, the capillary networks are less dense. The increasing demand during postnatal development of gray matter is reflected in the estimated doubling of capillary density [78]. Newly formed sprouting vessels are fragile and become stabilized by the recruitment of PVCs (such as pericytes, VSMCs, and astrocytes). A functional neurovascular unit is formed by the interactions between the invading ECs and the PVCs of the parenchyma [79,80]. In the CNS, retinal ganglion cells and astrocytes provide a physical template for sprouting ECs while releasing pro-angiogenic and anti-angiogenic factors such as VEGFA, semaphorins, and Nogo-A. Ablation of radial glia [81] or astroglia [82] results in a severe reduction in developmental angiogenesis [83].

In angiogenesis, the endothelial cell invasion contributes to the formation of a functional blood brain barrier (BBB) unit. The elements of an intact BBB include non-fenestrated endothelial cells and the tight junctions between them, the basal lamina, pericytes, astrocyte foot processes, and the perivascular space. As a physical barrier, tight junctions between endothelial cells and their lack of fenestrations serves to prevent solute penetration. The endothelial cells act as a continuous sheet of lipid bilayer that limits transfer of large molecules and water-soluble substances as dictated by the endothelial cell transport proteins. Thus, molecules are required to pass through several lipid bilayers to enter the brain parenchyma. The BBB functions to regulate the transport of micronutrients and macronutrients, receptor-mediated signaling, and leukocyte trafficking, and thus can selectively transport molecules [84]. Excellent reviews of the development and maturation of the BBB have been published [22,85,86,87]. Briefly, the tight junctions of the barrier are formed by GD 11 in the mouse and ED1 2 in the rat [88] and by gestational week 14 in the human [89]. The maturation of the density and complexity of the BBB continues postnatally [88,90]. Endothelial expression of the efflux transporter Pg-P (P8) [88] and ABC transporters [91] increases during postnatal development, indicating greater capacity for molecular efflux from the brain. The glucose transporter GLUT1 is expressed in early embryonic stages in capillary tight barrier structures, with changes in regional distribution occurring between postnatal days 3 and 60 [78,92]. While microglia are not a well-established component of the BBB, their association with capillaries in the early stages of development suggests a possible role in engulfing exogenous substances entering the pericapillary space via transendothelial transport [93]. Embryonic microglia that have been suggested to be critical for BBB angiogenesis are characterized by the presence of tyrosine kinase receptor (TIE2) and neuropilin 1 (NRP1) [94] and the expression of known macrophage markers such as the chemokine receptor, CX3CR1, CD11b/c, and CD45 by ED11 [95].

## 4. Microglia and Vascularization

In the mid-1930s, investigations examining the relationship between microglia and blood vessels were published. von Sa’ntha described the presence of microglia and their intimate association with blood vessels prior to mid-gestation in numerous non-primate species [96]. This work was expanded to examine this association in the rat embryo. It was found that the earliest appearance of microglia in the rhombencephalon and diencephalon coincided with the onset of vascularization [97]. Examination of the developing human brain showed similar patterns, with a temporal association of microglia progenitor influx occurring with vascularization [49]. It was not until the 1970s that the associations reported in these earlier studies were extended, finding precursors of ramified brain-resident microglia expressing the adult myeloid cell marker, F4/80, in the mature rat brain. These precursor cells were found in the vicinity of brain capillaries and characterized by a round or irregularly shaped cell body, with or without pseudopodia [98]. The colonization of the brain by microglia was observed to coincide with CNS vascularization during embryonic development [99,100]. In the fetal human brain microglia were located at highly vascularized sites at between 16- and 22-weeks gestation [101]. This coincided with the expression of intracellular adhesion molecule (ICAM)-2 in the cerebral endothelium, suggesting a contact relationship [101]. These human microglial progenitors showed a close association with the parenchymal wall of penetrating radial blood vessels [51]. It has been suggested that microglia serve an angiogenic role similar to that of macrophages, which includes the release of factors such as transforming growth factor beta (TGFβ), fibroblast growth factor 2 (FGF2), matrix metalloproteinases, and cytokines. It is thought that, as in the adult, these factors serve to break down the basement membrane directly or via stimulation of other cells [102,103,104] for the regulation of angiogenesis [105,106].

While there is likely a developmental interdependency between microglia colonization and vascularization, work by Fantin et al. [94] demonstrated that yolk sac-derived macrophages colonize the embryonic mouse brain in a manner that is independent of vessels; however, they rapidly associate with the sprouting vessels. The accumulation of microglia in the subventricular zone (SZ) between ED 10 and 11.5 occurs concurrently with lateral vessel sprouting and fusion to form the subventricular vascular plexus (SVP). The microglia number peaks in the SZ at ED 11.5 and declines at ED 12.5, correlating with the establishment of the SVP. The cells then begin to accumulate in deeper brain regions, suggesting a correlation with the formation of connections between neighboring radial vessels. As microglia mature, they extend processes to orientate toward and contact blood [107,108,109]. Throughout all phases of vascular network formation, microglia demonstrate some form of interaction with endothelial tip cells, and this interaction changes as the system becomes more complex as it matures [94]. In genetically modified mice carrying a loss-of-function mutation in the gene encoding the transcription factor PU.1, the deficiency in macrophages significantly decreased vessel intersections and network complexity [94]. A similar reduction in microglia complexity has been demonstrated in mice deficient in colony stimulating factor 1 (CSF-1) which manifests a deficiency in microglia [94]. In mice null for Vps35 (vacuolar protein sorting 35) in neurons, a reduction in vessel branching and density was observed, along with a decrease in microglia [110]. The additional depletion of microglia by PLX3397 exacerbated the vessel deficits [110]. Additional elegant exploration into the potential regulatory functions of microglia as compared to cells derived from circulating monocytes demonstrated that brain angiogenesis is promoted by cells of yolk sac origin and not by the circulating monocytes [94].

With neurovasculature establishment in the rat cerebrum and spinal cord, end-feet processes of juxtavascular microglia physically contact the basal lamina of microvessels [111]. Contact occurs with vessels of various sizes and types, including arteries, veins, and capillaries. A high percentage of juxtavascular microglia is associated with large capillaries in the early postnatal mouse cortex. This allows for migration along the vasculature during the peak time of colonization [112]. It has been demonstrated that, generally, microglia preferentially associate and contact the vasculature in areas that lack full astrocytic endfeet coverage [112]. Even though this specialized population of microglia resides in the parenchyma, they remain distinct from other parenchyma microglia according to the nature of their contact with blood vessels. The interaction between this microglia subpopulation and blood vessels is maintained throughout life and can be stimulated upon vessel injury. Using time-lapsed microscopy of brain slices from immature postnatal rodents, it was demonstrated that juxtavascular microglia easily migrate along the parenchymal surface of the vessel following traumatic stimulation [113]. In contrast, any stimulation of parenchyma microglia movement along blood vessels required a distinct retraction of microglia processes to accommodate the need for greater mobility. Grossmann et al. [113] reported that microglia along vessels display a heterogenous morphology. In general, they reported that microglia along vessels displayed a flattened morphology over the vessel surface, extending a leading process parallel to the vessel. Heterogeneity occurred, with some cells showing a transient response to stimulation while other cells maintained their migration for hours, resulting in movement over extended distances. Grossmann et al. also reported that a distinct group of blood vessel-associated microglia remained stationary and somewhat anchored to the blood vessel yet retained a level of motility by actively extending processes into the surrounding parenchymal tissue [113]. Figure 1 shows these two morphologies in a representative image of microglia associated with neurovasculature, providing a visualization.

The influence of brain vasculature on microglia can be observed in their morphological phenotype and development. During the normal maturation of microglia, cells show a transition shift from round non-process bearing cells that allow for ease of migration to a cell morphology characterized by extended processes before eventually maturing to cells expressing a fully ramified morphology. Under conditions of decreased capillary flow, the maturation of microglia is delayed, suggesting a direct influence on normal developmental processes [114]. In adults, microglia located in brain regions lacking a blood brain barrier, which include the circumventricular organs and Kolmer cells of the choroid plexus, display a morphological phenotype characterized by shorter, thicker processes [115]. Microglia located in brain areas receiving the best blood supply display a more complex process ramification pattern than the stunted morphology seen in areas lacking a blood brain barrier (BBB) [116,117]. It is thought that this difference in morphology could reflect the influence of serum proteins made available to the cells due to diminished BBB integrity [118].

In the avian embryo, the use of chick-quail transplantation and parabiosis chimeras suggested that vascularization was not required for microglial colonization of the CNS. Cells were observed to invade the CNS through the pial basal lamina before vascularization occurred [119,120]. In the zebrafish embryo, live recordings of cell movement showed migration of microglia through the cephalic mesenchyme toward the brain pial surface and roof of the fourth ventricle, with colonization of the brain and retina occurring through a macrophage colony stimulating factor-1 (M-CSF-1) receptor-dependent process [121].

## 5. Microglia and Vascularization in the Spinal Cord

In the embryonic mouse spinal cord, microglia expressing a fluorescent marker were identified as early as ED 11.5 inside the primitive arterial tract. Between ED 12.5 and ED 14.5, these cells proliferated and accumulated in the dorsal part of the perineural vascular plexus (PNVP) [122]. Within the external dorsolateral region of the spinal cord, microglia showing an amoeboid morphology clustered close to terminals of dying dorsal root ganglia neurons. They were also observed to cluster within the lateral motor columns at the onset of developmental motoneuron programmed death. Between ED 12.5 and ED 13.5, ramified microglia within the parenchyma were observed to interact with growing capillaries. At ED 14.5, approximately 30% of the total microglia within the gray matter had initiated contact with growing capillaries. While microglia were evident in the surrounding area, further examination of the immunoreactivity of the blood vessel using von Willebrand factor along with fluorescent cell bodies in the gray matter suggested that the contact of microglia with developing blood capillaries in the spinal cord did not reflect a route for colonization [122].

## 6. Microglia and Vascularization in the Retina

The most prominent data examining the relationship between microglia colonization and vascularization are provided through examination of the retina [101,123,124,125,126]. In the retina, heterogeneous populations of microglia are believed to be of hemangioblast mesodermal origin [127], with yolk sac derived microglia invading the eye during embryogenesis. Vessel sprouts emerge from the optic nerve head and spread perpendicularly over the retinal surface. The sprouting continues downwards into the inner layers of the retina, where sprouts fuse to form the intermediate plexus. It is thought that this occurs in the mouse around the second week after birth and that the deep vascular plexus innervation occurs in the third week [128]. The cells first populate the superficial layer of the retina and then migrate into the deeper layers, thereby experiencing significant changes in tissue stiffness [129]. In the mouse, the endothelial tip cells at the sprouting front of the retinal vasculature were shown to be in close contact with ramified microglia that could bridge sprouts while they were anastomosing [89]. While the microglia appeared more differentiated, interactions with endothelial cells were consistent with what has been observed in the developing hindbrain.

Early research characterized a heterogeneous population of microglia in the adult human retina. The developmental distribution of these cells suggested that cells bearing macrophage markers were ontogenetically distinct from microglia that did not display such markers [130]. In this series of studies, it was suggested that microglia enter the retina from the ciliary margin prior to vascularization and from both the optic disc and ciliary margin after vascularization. The unique macrophage antigen positive microglia were considered to enter the retina mainly via the optic nerve head [130]. During human retinal development, microglia that lack macrophage markers were present by gestational week 10. This occurred prior to astrocyte invasion and the onset of vasculogenesis [131]. While macrophage antigen expressing microglia (CD45, MHC-I, and MHC-II) could be observed in the retina prior to vascularization, they predominantly arrived along with the vascular precursors at approximately gestational weeks 14–15 [131,132]. It was thought that the macrophage antigen expressing cells may represent vessel-associated perivascular microglia as detected in the adult retina and that the MHC positive, but macrophage-antigen negative, microglia mature to the ramified microglia seen in the adult retina [130,132]. The work of Fantin et al. [94] demonstrated that the major population of tissue macrophages at the time of brain vascularization are yolk sac-derived macrophages expressing the transmembrane proteins essential for angiogenesis, such as the angiopoietin receptor TIE2 and the NRP1 protein that modulates intercellular adhesion. These cells interacted with endothelial tip cells and promoted vascular anastomosis downstream of VEGF-mediated tip cell formation and sprout induction. Fate mapping and flow cytometry analysis further identified the unique phenotype of adult retinal microglia with a unique CD45^low/^CD11c^low/^F4/80^low/^I-A/I-E^-^ signature. Interestingly, this signature remains conserved in the steady state and during retinal injury [133].

With vascular development, microglia migrate through layers of increasing stiffness in the retina and undergo polarization by assuming a bipolar rod shape in the stiffest outer nuclear layer [129]. Establishment of appropriate retinal blood vessel formation associates with an adequate resident microglial population. Depletion of resident retinal microglia using clodronate liposomes reduced vessel growth and density [101]. However, depletion of microglia by the downregulation of colony stimulating factor 1 (CSF-1) diminished branching anastomosis during retinal development without altering the number of endothelial tip cells and filopodia [134]. A diminished number of vascular branch points was also observed in the hindbrain after CSF-1-dependent depletion of microglia [89]. The absence of microglia altered the angle of filopodia extending from tip cells [135]. Clodronate liposome depletion of microglia in ex vivo retinas significantly reduced the potency of VEGF-induced neovascular sprouting [136].

Cell-cell communication between microglia and the vasculature in the retina has been demonstrated. Notch1 (neurogenic locus notch homolog protein 1)-activated retinal microglia were found in close association with Notch ligand and Delta-like ligand 4 (Dll4)-expressing endothelial tip cells [137]. With genetic deletion of Notch1 in retinal microglia, the number of microglia localized to the vasculature was decreased. The process of vessel sprouting in cultures of the aortic ring has been demonstrated to be influenced by microglia contact and communication via soluble microglia-derived factors. In communication between microglia and aortic rings, neutralizing VEGFR activation did not alter the induction of branching [135]. Retinal myeloid cells produce Wnt ligands that suppress angiogenic branching through a non-canonical Wnt-Flt1 pathway [138]. While canonical Wnt signaling has been reported to regulate assembly of CNS vasculature [139], haploinsufficiency of Wnt5a, Wnt11, the common Wnt-ligand transporter Wls, or the VEGF inhibitory protein sFlt1, increased vascular branching in the deep vascular plexus of the retina. However, the lack or absence of microglia had no effect on branch density in this region [89]. Further evidence is available supporting an interaction between microglia and the vasculature during colonization of the retina in the form of an evaluation of mice with a genetic deletion of the angiotensin receptor MAS (Mas1^−/−^). In Mas1^−/−^ mice, a reduced number of microglia was observed, along with reduced progression of the vascular front and decreased vascular density as compared to wildtype mice [140].

## 7. Conclusions

While there is a wealth of reviews on microglia in the literature, there are few that target the developmental aspect of their interaction and possible interdependency with neurovascular development. The focus of this review has been targeted to facilitate our current understanding of the ontogeny of microglia and neurovascularization and the potential early and critical interactions between microglia and vessels. In that regard, and in consideration of the fact that the complex processes involved require an integrated whole animal system, the cited literature relied on in vivo studies only and did not speculate on possible supportive data obtained from in vitro studies. The currently available data provided information on the timing of the presence of microglia in the brain and their shifting morphology, as well as information on the presence of vascular components. What is missing are detailed examinations of the interdependency between the two and a better understanding as to how the relationship is similar to, or different than, what is known in the adult brain. Many of the assumptions surrounding developmental interactions have been derived from studies examining the formation of the retina. Whether these reflect dynamics in the brain or spinal cord is not yet known, but it is evident that there are some differences. There is no debate as to the essential nature of these processes; however, how these processes interact with and influence each other remains to be characterized. The developmental timing and location of microglia in relationship with the brain vasculature, as well as radial glia and progenitor cells, imply close and functional associations. These interactions and associations comprise physical cell contact as well as the presentation of critical signaling factors, including pro-gliogenic and pro-neurogenic molecules. While many of these cellular components may become evident in the brain following microglia colonization, how they serve to focus the cell interactions to ensure successful development remains an investigative question.

The increase in studies examining microglia during development and the various roles they undertake to perform a vast array of beneficial tasks critical for forming an intact and functioning neural network should allow for increased understanding of associated dynamics and regulatory processes. Further efforts are needed to understand the interactive relationship between the very early processes of microglia colonization and vascular innervation. The recent establishment and use of techniques such as light-sheet microscopy and spatial transcriptomics may allow for more integrated and sensitive approaches to interrogate developmental processes and help to characterize critical cell-cell interactions. It is anticipated that such work will provide a basis for understanding how genetic factors, maternal health and disease state, and various types of environmental, pharmaceutical, and stress-related factors may alter the early gestational environment in a manner that can lead to adverse health outcomes.

## Figures and Tables

**Figure 1 ijms-25-01281-f001:**
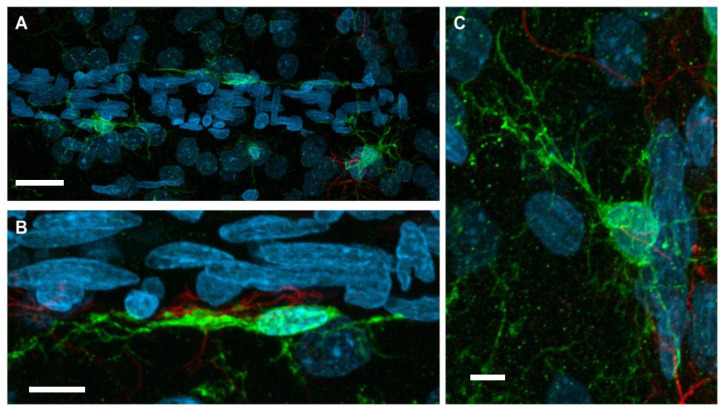
Representative con-focal microscopy(40x/1.3 na oil objective; total mag 400x at a resolution of 0.26 μm) images of Iba-1^+^ microglia in contact with vasculature of the male Sprague-Dawley adult rat cortex. Immunofluorescent staining was conducted on 30 micron sections of 4% paraformaldehyde fixed brains (FD Neurotechnologies, Columbia, MD, USA). (**A**) Image of Iba-1^+^ microglia (green) adjacent to the vasculature (flat, elongated endothelial cells (DAPI—blue)). (scale bar 20 μm) (**B**,**C**) At higher magnification the two distinct types of morphology for microglia in contact with vasculature are presented. Scale bar 5 μm. (**B**) An elongated microglia cell body and processes along the vasculature or (**C**) microglia with processes that contact the vasculature and extend processes into the parenchyma for contact with neurons. Red fluorescence represents GFAP^+^ astrocytes. Images were collected as part of an ongoing research effort in the author’s laboratory and are only included as a visualization of the two types of microglia morphologies along the vasculature.

## Data Availability

Not applicable.

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
