# Peer review of "Developmental Associations between Neurovascularization and Microglia Colonization"

_ijms, 2024, doi:10.3390/ijms25021281_

Round 1
Reviewer 1 Report
Comments and Suggestions for Authors
This article presents data clearly and it represents a scientific contribution because it focuses on the developmental ontogeny and interdependency between the colonization of the nervous system with yolk-sac derived macrophages and vascularization. Abstract and title are informative and reflect the content of the paper, appropriate keywords have been given.
The introduction contains well-selected and cited literature, and is sufficient to explain the topics of the paper. Perhaps even too extensive in relation to the conclusion.
This part from line 252 is written a little confusingly. It is not clear whether the author did these experiments himself. If it is his work, that part should be better described, how, when and how many samples. All the details. Or not to state at all. Because it's confusing if you mix other people's results with your own, and everything is stated the same.
4. Microglia and vascularisation in spinal cord and 5. Microglia and vascularisation in retina should be subheaders of 3. Microglia and vascularisation.
Conclusions are justified by the data presented but I would expect a little more about what is not known yet. In my opinion it would be better to first highlight and emphasize a bit more the most important things that are known, then explain what still needs to be researched. Finally, I would like you to emphasize why this review is important. Furthermore, the strengths and limitations of this review are not highlighted.
Finally, there are also some grammar corrections needed (in line 22, 28,…)
Comments on the Quality of English LanguageThe English language needs improvement, it has a significant number of typos and occasional grammatical and sentence errors
Author Response
Figure 1 has been clarified as having been captured in my lab - this is in the figure legend and the text has been edited to better reflect that it is not from a cited article.
"
Grossmann et. al. [113] reported that microglia along vessels display a heterogenous morphology. In general, they reported that microglia along vessels displayed a flattened morphology over the vessel surface, extending a leading process parallel to the vessel. A heterogeneity occurred with some cells showing a transient response to stimulation while other cells maintained their migration for hours, resulting in movement over extended distances. Grossmann et al [113] also reported that a distinct group of blood vessel-associated microglia remained stationary and somewhat anchored to the blood vessel yet, retaining a level of motility by actively extending processes into the surrounding parenchymal tissue [113]. Figure 1 shows these two morphologies in a representative image of microglia associated with the neurovasculature, providing a visualization."
moved to section 3.
Newly formed sprouting vessels are fragile and become stabilized by the recruitment of PVCs (such as pericytes, VSMCs and astrocytes). A functional neurovascular unit is formed by the interactions between the invading ECs and their interactions of PVCs of the parenchyma [79,80]. In the CNS, retinal ganglion cells and astrocytes provide a physical template for sprouting ECs while releasing pro-angiogenic and anti-angiogenic factors such as VEGFA, semaphorins and Nogo-A. Ablation of radial glia [81] or astroglia) [82] results in a severe reduction in developmental angiogenesis [83].
The conclusion has been edited
"
While there is a wealth of reviews on microglia in the literature, there are few that target the developmental aspect of their interaction and possible interdependency with neurovascular development. The focus of this review has been targeted to our current understanding of the ontogeny of microglia and neurovascularization and the potential early and critical interactions between microglia and vessels. In that regard, and the consideration that the complex process required an integrated whole animal system, the cited literature relied on in vivo studies only and did not speculate on possible supportive data obtained from in vitro studies. The currently available data provided information on the timing of the presence of microglia in the brain and their shifting morphology as well as information on the presence of vascular components. What is missing are detailed examinations of the interdependency between the two and a better understanding as to how the relationship is similar or different than what is known in the adult brain. Much of the assumptions for developmental interactions have been derived from studies examining the formation of the retina. Whether that reflects dynamics in the brain or spinal cord are not yet known but it is evident that there are some differences. There is no debate as to the essential nature of these processes however, how these processes interact and influence each other remains to be characterized. The developmental timing and location of microglia in relationship with the brain vasculature, as well as radial glia and progenitor cells, imply close and functional associations. The interactions are comprised of physical cell contact as well as presentation of critical signaling factors including pro-gliogenic and pro-neurogenic molecules. While many of these cellular components may become evident in the brain following microglia colonization how they serve to focus the cell interactions to ensure successful development remains an investigative question.
The increase in studies examining microglia during development and the various roles they undertake to perform a vast array of beneficial tasks critical for forming an intact and functioning neural network should allow for increased understanding of associated dynamics and regulatory processes. Further efforts are needed to understand the interactive relationship between the very early processes of microglia colonization and vascular innervation. The recent establishment and use of techniques such as light-sheet microscopy and spatial transcriptomics may allow for more integrated and sensitive approaches to interrogate developmental processes and help to characterize critical cell-cell interactions. It is anticipated that such work will provide a basis for understanding how genetic factors, maternal health and disease state, and various types of environmental, pharmaceutical, or stress-related factors may alter the early gestational environment in a manner that can lead to adverse health outcomes. "
Points of either typos or verb structure have been edited
The order of the sections has been maintained as trying to sub-topic sections was awkward when I tried to edit.
Reviewer 2 Report
Comments and Suggestions for Authors
This review focuses on the interdependent relationship between microglia colonization and vascularization in the early stages of nervous system development. It specifically examines the role of yolk-sac derived macrophages and their co-development with the nervous system's vascular network. Emphasizing the importance of understanding the timing and mutual influence of these processes, the review highlights their implications for interpreting alterations during early development. It also discusses how these developmental aspects are affected by the gestational environment, genetics, and external factors, and their potential long-term impacts on the nervous system.
The paper is well-written and the evidence is laid out clearly.
Author Response
I thank the reviewer for appreciating the very focused nature of this review and the importance of trying to understand the interactions during development outside of simply applying those that may occur in the adult.
The conclusions have been edited to address comments of Reviewer 1
Conclusion
While there is a wealth of reviews on microglia in the literature, there are few that target the developmental aspect of their interaction and possible interdependency with neurovascular development. The focus of this review has been targeted to our current understanding of the ontogeny of microglia and neurovascularization and the potential early and critical interactions between microglia and vessels. In that regard, and the consideration that the complex process required an integrated whole animal system, the cited literature relied on in vivo studies only and did not speculate on possible supportive data obtained from in vitro studies. The currently available data provided information on the timing of the presence of microglia in the brain and their shifting morphology as well as information on the presence of vascular components. What is missing are detailed examinations of the interdependency between the two and a better understanding as to how the relationship is similar or different than what is known in the adult brain. Much of the assumptions for developmental interactions have been derived from studies examining the formation of the retina. Whether that reflects dynamics in the brain or spinal cord are not yet known but it is evident that there are some differences. There is no debate as to the essential nature of these processes however, how these processes interact and influence each other remains to be characterized. The developmental timing and location of microglia in relationship with the brain vasculature, as well as radial glia and progenitor cells, imply close and functional associations. The interactions are comprised of physical cell contact as well as presentation of critical signaling factors including pro-gliogenic and pro-neurogenic molecules. While many of these cellular components may become evident in the brain following microglia colonization how they serve to focus the cell interactions to ensure successful development remains an investigative question.
The increase in studies examining microglia during development and the various roles they undertake to perform a vast array of beneficial tasks critical for forming an intact and functioning neural network should allow for increased understanding of associated dynamics and regulatory processes. Further efforts are needed to understand the interactive relationship between the very early processes of microglia colonization and vascular innervation. The recent establishment and use of techniques such as light-sheet microscopy and spatial transcriptomics may allow for more integrated and sensitive approaches to interrogate developmental processes and help to characterize critical cell-cell interactions. It is anticipated that such work will provide a basis for understanding how genetic factors, maternal health and disease state, and various types of environmental, pharmaceutical, or stress-related factors may alter the early gestational environment in a manner that can lead to adverse health outcomes.
Reviewer 3 Report
Comments and Suggestions for Authors
The studies on roles of microglia have been extensively progressed in neurodegeneration such as Alzheimer's disease and Parkinson's disease, nervous system injury and repair, and aspects of neural circuitry refinement. However, understanding of the immature microglia remains less clear.
This review article precisely overviews the findings on microglia especially in termes of interdependency between immature microglia colonization and vascularization. The migration of microglia progenitors and their entry into the brain parenchima are well described by following the historical literatures.
There are some minor typographical errors: 1) Numbers of sections are wrong. Number 2 sections are duplicated, so the numbers of the sections after second number 2 should be changed: 2 -->3; 3 --> 4; 4 --> 5; 5 --> 6; 6 --> 7; 2) Line 210: transforming growth factor beta (TGF beta); 3) Line 295: PVCs (such as pericytes, VSMS, and astrocytes); 4) Reference [108] should be cited in the legend of Fig 1, and the copyright should be clarified for Fig.1.
Author Response
I appreciate the effort of the reviewer in catching those things that get past one. The typographical errors, I think, have been corrected.
The information on Figure 1 is included and the paragraph referring the Grossmann et al has been clarified so there is no confusion that Figure 1 represents his work. This was to serve as a representative image of the two types of cells described by 108 - but not from that reference. The images were captured in my laboratory. I have edited the figure legend and the paragraph above to try to more clearly state this.
Figure 1. Representative con-focal images of Iba-1+ microglia in contact with vasculature of the male Sprague-Dawley adult rat cortex. Immunofluorsecent staining was conducted on 30 micron sections of 4% paraformaldehyde fixed brains (FD Neurotechnologies, Columbia, MD, USA). (A) Iba-1+ microglia (green) adjacent to the vasculature (flat, elongated endothelial cells (DAPI - blue)). Microglia in contact with the vasculature display two distinct types of morphology. (B) An elongated cell body and processes next to the vasculature or (C) microglia with processes that contact the vasculature and extends processes into the parenchyma for contact with neurons. Red fluorescence represents GFAP+ astrocytes. Images were collected as part of an ongoing research effort in the author’s laboratory and are only included as a visualization of the two types of microglia morphologies along the vasculature.
The conclusions have been edited to address comments of Reviewer 1
Conclusion
While there is a wealth of reviews on microglia in the literature, there are few that target the developmental aspect of their interaction and possible interdependency with neurovascular development. The focus of this review has been targeted to our current understanding of the ontogeny of microglia and neurovascularization and the potential early and critical interactions between microglia and vessels. In that regard, and the consideration that the complex process required an integrated whole animal system, the cited literature relied on in vivo studies only and did not speculate on possible supportive data obtained from in vitro studies. The currently available data provided information on the timing of the presence of microglia in the brain and their shifting morphology as well as information on the presence of vascular components. What is missing are detailed examinations of the interdependency between the two and a better understanding as to how the relationship is similar or different than what is known in the adult brain. Much of the assumptions for developmental interactions have been derived from studies examining the formation of the retina. Whether that reflects dynamics in the brain or spinal cord are not yet known but it is evident that there are some differences. There is no debate as to the essential nature of these processes however, how these processes interact and influence each other remains to be characterized. The developmental timing and location of microglia in relationship with the brain vasculature, as well as radial glia and progenitor cells, imply close and functional associations. The interactions are comprised of physical cell contact as well as presentation of critical signaling factors including pro-gliogenic and pro-neurogenic molecules. While many of these cellular components may become evident in the brain following microglia colonization how they serve to focus the cell interactions to ensure successful development remains an investigative question.
The increase in studies examining microglia during development and the various roles they undertake to perform a vast array of beneficial tasks critical for forming an intact and functioning neural network should allow for increased understanding of associated dynamics and regulatory processes. Further efforts are needed to understand the interactive relationship between the very early processes of microglia colonization and vascular innervation. The recent establishment and use of techniques such as light-sheet microscopy and spatial transcriptomics may allow for more integrated and sensitive approaches to interrogate developmental processes and help to characterize critical cell-cell interactions. It is anticipated that such work will provide a basis for understanding how genetic factors, maternal health and disease state, and various types of environmental, pharmaceutical, or stress-related factors may alter the early gestational environment in a manner that can lead to adverse health outcomes.